# Training the Trainers in Language Assessment via Mentoring: Building Expertise to Promote Language Assessment Literacy of Ukrainian University Teachers

**Viktoriya Osidak \*, Olha Drahinda \* and Olga Kvasova \***

Department of Methods of Teaching Ukrainian and Foreign Languages and Literature, Taras Shevchenko National University of Kyiv, 016016 Kyiv, Ukraine
\* Correspondence: V.Osidak@mail.knu.ua (V.O.); o.drahinda@mail.knu.ua (O.D.); o.kvasova@mail.knu.ua (O.K.)

**Abstract:** This mixed-methods study described a case of group-based informal mentoring, a conceptual model of which was aligned with an effective mentoring program found in the literature. The research questions that were addressed in the study included: (1) Will mentoring training, conducted within a short (1-month) period, enable inexperienced presenters to develop and conduct an effective workshop in LTA? (2) In what way(s) has mentoring training impacted the mentees as prospective trainers in LTA? The training was implemented by one mentor and two mentees, with the immediate purpose to organize a platform to disseminate the results of an Erasmus+ staff mobility program. The data were collected via a questionnaire, a self-efficacy scale and reflection logs about the mentoring procedure, as well as the workshop in LTA conducted by the mentees with the purpose to enhance Ukrainian university teachers' LAL. The outcome of the training reported a high success level of the workshop among 37 attendees. In addition, the participants of the mentoring training (*n* = 2) demonstrated improvement in their organizational skills and professional growth. The mentoring framework proved to be an effective strategy for implementing study objectives and can be recognized as a successful model for the promotion of language assessment literacy.

**Keywords:** language assessment literacy; mentoring; mentor and mentees; workshop construct; teacher profile

## 1. Introduction

With language assessment literacy (LAL) gaining significance for all stakeholders lately (Pill and Harding 2013; Taylor 2013), the need for training teachers in conducting fair assessments has become critical (Vogt et al. 2020). Successful implementation of Massive Open Online Courses (e.g., Language Assessment in the Classroom, the British Council) along with the accomplishment of eight online courses developed by the TALE project (Tsagari et al. 2018) have broadened and increased the number of participants from across the world. The pandemic-caused transition of in-person conferences and training events in language testing and assessment (LTA) to the online mode has offered invaluable opportunities for grassroots teachers to learn from world-acclaimed experts whose free webinars became available to them.

Giving due credit to the effort of the LTA community to share expertise with the teachers in these difficult times, it is desirable that teacher training in language assessment be contextualized in line with the national/local requirements and educational tradition. In Ukraine, according to Kvasova's (2018) surveys, university teachers preferred face-to-face training events to online courses and webinars explaining this preference by the need for immediate feedback on their progress from trainers. Acknowledging webinars as an effective means of developing LAL, in response to another survey, teachers expressed their favor of taking short-term courses conducted by local trainers (Kvasova and Shovkovy 2020). Although both surveys were held before the pandemic, the data should be taken

into account while planning further stages of LAL development. For Ukrainian context, such next stage is associated with the organization of short-term courses in LTA, which suggests that a team of trainers should be prepared to conduct workshops on the aspects included in the course curriculum.

Responding to the above needs, this study aimed to present the procedure and outcomes of training the trainers in LTA. We chose mentoring for such training since this framework proves effective in nurturing less experienced colleagues, enhancing their professional development through the support and assistance of a knowledgeable and competent mentor (Greiner et al. 2017; Skjevik et al. 2020; Whitehouse 2016). Moreover, since Ukrainian higher education has not run graduation programs in LTA up to now, the mentors, who have received training participating in international projects, are viewed as a key source of expertise in the field. Therefore, mentoring appears an almost sole way to train the trainers in LTA, with a mentor being an assurer of novice trainers' ability to further pass on LTA skills to practicing teachers.

## 2. Literature Review

### 2.1. Language Teacher LAL Training Needs Analysis

According to the definitions of LAL offered in literature, LAL can be described as a repertoire of competences or skills and specific levels of knowledge (Bøhn and Tsagari 2021; Coombe et al. 2020; Taylor 2013; Vogt and Tsagari 2014) that enable an individual to understand, assess, construct language tests and analyze and interpret test data effectively, using relevant assessment methods. LAL plays a crucial role in language teacher qualification and has a part in many classroom activities. As cited in Giraldo (2018), teachers who have received training in LAL and demonstrate a high level of LAL use assessment practices to enhance teachers' instruction and their students' learning and contribute to the development of learner autonomy. A study by Hakim (2015) showed a correlation between teacher experience and assessment practices used in the classroom. Remarkably, though, multiple research studies report an insufficient level of teachers' LAL development (Bøhn and Tsagari 2021; Drackert and Wolfgang 2018; Coombe et al. 2020; Giraldo 2018; Kvasova and Kavytska 2014; Taylor 2013; Vogt and Tsagari 2014).

The findings of the study carried out by Vogt and Tsagari (2014) with teachers in European countries indicate an overall need for teacher training across a variety of aspects of the LTA literacy frame despite the fact that the LAL level has improved in the last decade. The majority of the participants of the study had problems with the identification of "purposes of training" and expressed needs for more advanced training for the assessment of "receptive" and "productive skills" as well as for the "microlinguistic aspects" and "integrated skills". In addition, many teachers reported insufficient training in the field of establishing "validity" and "reliability" of assessment results and using "statistics". Moreover, Vogt and Tsagari (2014) found that language teachers were not confident about how to implement innovative forms of LAL such as self- and peer assessment and portfolios in their classroom practices despite the fact that they supported using alternative forms of assessment. Another interesting finding of the study by Hasselgreen et al. (2004) was that the majority of teachers used assessment practices without any formal training in the field. Very similar LTA problems and training needs were outlined in Şişman and Büyükkarcı's analysis of research data on teacher assessment needs in Turkey (2019). Moreover, Drackert and Wolfgang (2018) and Şişman and Büyükkarcı (2019) reported that teachers in Austria, Switzerland, Germany and Turkey admitted giving a personal interpretation of such fundamental assessment concepts as "validity" and "reliability" and, consequently, misused them in practice. Other studies revealed that teachers do not know how to interpret and communicate assessment data to promote students' learning (Kleinsasser 2005) or use summative assessment for classroom-based assessment (DeLuca and Klinger 2010). Yan et al. (2018), who attempted to specify important factors of language assessment literacy development for secondary-level Chinese teachers through semi-structured retrospective interviews, concluded that teachers had a distinct language

assessment literacy profile and required more formal training in assessment practices than in theories of assessment. Moving to Ukrainian context, the results of a study on the writing assessment literacy of university language teachers demonstrate the need for an improvement in writing assessment practices, which could be achieved through providing training and reorientation to help Ukrainian teachers develop common understanding and interpretation of task requirements and scale features (Kvasova et al. 2019). In general, research findings on teacher LAL needs collected in different countries (Bøhn and Tsagari 2021; Coombe et al. 2020; Drackert and Wolfgang 2018; Kvasova et al. 2019; Mahapatra 2016; Montee et al. 2013; Şişman and Büyükkarcı 2019; Vogt et al. 2020) call for more rigorous and advanced training in LAL as a part of teachers' professional development.

There are numerous studies that specifically focused on teacher perceived professional needs in LAL development which put assessment training in the spotlight (Yan et al. 2018; Şişman and Büyükkarcı 2019; Vogt and Tsagari 2014). The majority of teachers in the studies underwent some pre- or in-service training at some point in their careers. However, no evidence was provided that the training received covered all aspects of LTA literacy necessary for classroom needs (Coombe et al. 2020; Drackert and Wolfgang 2018; Vogt and Tsagari 2014). In this regard, summing up the analysis of the related research data, Coombe et al. concluded that "most training programs only include a generic assessment course which provides insufficient detail for developing an adequate assessment knowledge base" (Coombe et al. 2020). Interestingly, teachers in Drackert and Wolfgang's (2018) study did not express a need for training in LAL despite the fact they recognized the importance of LAL for language teacher professional development. Yet, insufficient or inadequate training may result in inappropriate use of assessment practices (Coombe et al. 2020; Kvasova and Kavytska 2014; Şişman and Büyükkarcı 2019) or teachers may avoid incorporating assessment strategies overall if their first experience was negative (Vogt and Tsagari 2014).

Customarily, teachers try to bridge the gap in assessment practices by learning from other colleagues or their mentors (Drackert and Wolfgang 2018; Vogt and Tsagari 2014). In this regard, Vogt, Tsagari and Spanoudis concluded that "collaboration between teachers seems to be crucial to compensate a lack of formal training and to build up a practical base of skills related to LAL" (Vogt et al. 2020, p. 402). However, different studies advocate the significance of online and face-to-face training in LTA literacy, which, as findings demonstrate, contributes to language teachers' professional development (Mahapatra 2016; Montee et al. 2013). For example, short-term face-to-face LAL programs positively affect teachers' perception of assessment and encourage them to make assessment practices a part of their classroom activities (Kvasova and Shovkovy 2020; Montee et al. 2013). In addition, online assessment training helped teachers improve their LAL levels (Mahapatra 2016). Universally, timely and continuous training conducted by experts has been long recognized as an effective strategy for supporting teacher assessment needs and also introducing teachers to the most recent and relevant assessment practices (for example, to alternative forms of assessment that are increasingly important nowadays and to practices used for assessment for, of and as learning).

### 2.2. Mentoring Defined

In the most general sense, mentoring is defined as a process of nurturing a less experienced colleague with the purpose of enhancing his/her professional development in the desired field through the support and assistance of a more experienced professional grounded in a theoretical and practical framework (Luckey 2009; Brown et al. 2020; Whitehouse 2016). Related literature review indicates that opportunities that mentoring offers for professional development significantly affect efficacy levels of personal and professional growth (Luckey 2009; Schaefer 2010; Woolfolk and Burke-Spero 2005). Among various types of induction support—personal, social and professional—that novice teachers can have, mentoring is viewed as the most conducive to their professional development (Greiner et al. 2017). Moreover, mentoring has gained popularity within organizations as an educational and dissemination strategy that supports collaborative training, education

and learning (Blake 2016; Brown et al. 2020; Hofmann and Springer 2014; Greiner et al. 2017; Risner et al. 2020; Skjevik et al. 2020; Whitehouse 2016). Therefore, many institutions tend to rely heavily on mentoring within professional and discipline frameworks to assist in the achievement of desirable professional outcomes or to support novices in their professional development (Allen et al. 2006; Blake 2016; Brown et al. 2020; Whitehouse 2016).

### 2.2.1. Theoretical Framework for Mentoring

For the purpose of this study, we used several developmental and learning theories that lay the foundation for the development of the concept of this mentoring model: self-efficacy and social cognitive theories, social cognitive career theory, adult development theory and productive mentoring (Bachkirova 2011; Bandura 1986, 2006; Brown et al. 2020; Connolly et al. 2018; Kram 1985; Varghese and Finkelstein 2020). Self-efficacy theory draws on the correlation between an individual's experience of perceived levels of productivity and their successful performance in selected surroundings. In the most general sense, high self-efficacy beliefs increase confidence and motivate an individual to work hard in order to reach success, and on the contrary, individuals with low self-efficacy levels tend to fail their commitments (Woolfolk and Burke-Spero 2005). Social cognitive theory adds that, in order to gain desirable outcomes, individuals build on their knowledge and behavioral skills. Within this model, the mentor is a key figure in mentoring training, regarded as a source of knowledge and expertise (Bandura 1986; Brown et al. 2020). Social cognitive career theory framework has been used as a university platform for teacher development and mentoring at a doctoral level. This framework is based on individuals' interests and efficacy beliefs and is targeted at increasing their knowledge and skills that lead to desirable outcomes (Bandura 1986, 2006; Brown et al. 2020). According to adult development theory, adult learners in any educational program would benefit from mentors who have experienced similar paths to their development and with whom they can establish a trustworthy and informal relationship (Kram 1985) "that represent a mixture of a parent and a peer" (as cited from Brown et al. 2020, p. 23). For example, novice teachers indicated that, among the most significant features that impacted their mentoring experience, the greatest were time spent with the mentor, communication, quality of the relationship and support from the mentor. Furthermore, a mentor's nonjudgmental feedback, guidance and opportunity for professional growth became decisive factors for novice teachers to stay in the profession (Hofmann and Springer 2014). The idea of productive mentoring is based on a developmental model that explains the stages of personal and professional growth from a novice to an expert in the field (Simmie and Moles 2011). This framework also supports the idea of equal personal, academic and professional development of both the mentor and mentees.

### 2.2.2. Mentoring Models

Among a variety of conceptual models for mentoring, two distinct models—informal and formal mentoring—are singled out. Both of the models have their advantages and disadvantages. Formal mentoring is assigned by third parties. As a result, it is easier to establish accountable standards (Allen et al. 2006; Blake 2016; Luckey 2009; Risner et al. 2020). In addition, formal mentoring is defined as a cooperative and discursive process. As it focuses on the dialogue between a mentor and his protégé, formal mentoring provides opportunities for reflective development. Such interaction is expected to result in the professional growth of both the mentor and their protégé. In other words, formal mentoring integrates involvement and crossover effects of the parties, joint planning and cooperation while carrying out the task (Allen et al. 2006; Awaya et al. 2003; Luckey 2009). On the other hand, in the formal modality, the mentor–mentee relationship usually lacks personal compatibility and connection, which may be one of the reasons for low commitment (Blake 2016; Whitehouse 2016).

Research data collected by Allen et al. (2006) and Blake (2016) demonstrated that mentoring in many professional fields is initially informal and can be defined as "extension

of another work-related role" (Allen et al. 2006, p. 5) or as "pastoral guidance, educational support or professional role modelling" (Blake 2016, p. 6). Consequently, an informal mentoring relationship is formed spontaneously and based on personal commonalities or shared commitments or preferences and likings. Yet, the informal model is characterized as one-way development. This means that, during the training of novice presenters, the mentor's input in the form of shared knowledge and expertise, support and guidance enables their protégés to reach the level of expertise and competence of the mentor. The mentor's professional development remains, though, unchanged, as informal training does not require reflection (Whitehouse 2016). At the same time, some study findings provide evidence that informal training can produce more impressive outcomes than formal training (Allen et al. 2006). Many formal programs attempt at mimicking the closeness of interpersonal processes underlying informal mentoring to facilitate formal mentoring (Allen et al. 2006; Ragins et al. 2000).

Common elements for the implementation of mentoring are briefings, practical interventions, in-field observations, consultations, etc. (Awaya et al. 2003; Blake 2016; Cox 2003; Resta et al. 2013; Skjevik et al. 2020). A simple orientation for a successful mentoring relationship may suffice in one context. In other cases, regular meetings or interventions are necessary. Moreover, physical proximity and frequency of the mentoring relationship, belonging to the same professional field and mentor's better rank or power position that can serve as a role model are important aspects for successful program outcomes (Allen et al. 2006; Skjevik et al. 2020). For these reasons, "matching mentors and protégés from the same department" may foster interaction frequency, role modeling and psychosocial support (Allen et al. 2006, p. 575). As a result, the effectiveness of mentoring can increase.

Regardless of the implementation styles, Resta et al. (2013) and Whitehouse (2016) stressed the importance of high-quality principles upon which any mentoring program should be based, such as "continuous inquiry into practice, self-assessment, reflection, mentor's responsive support tailored to the needs of a mentee" (Whitehouse 2016, p. 21). Literature analysis also provides evidence that in order to be successful and effective a mentoring program should be focused and structured with clearly defined sets of goals (Allen et al. 2006; Whitehouse 2016).

## 3. Defining the Goals of the Workshop

Coombe et al. (2020) concluded in their review of literature on assessment literacy that "assessment literacy should be developed by considering various educational contexts and necessities of times and context" (Coombe et al. 2020, p. 11). With this in mind and in order to specify the workshop LAL construct that can be relevant for Ukrainian language teachers' LAL needs and expectations, we focused on Taylor's (2013) LAL program, which provides a differentiation profile of different stakeholders' needs (Figure 1).

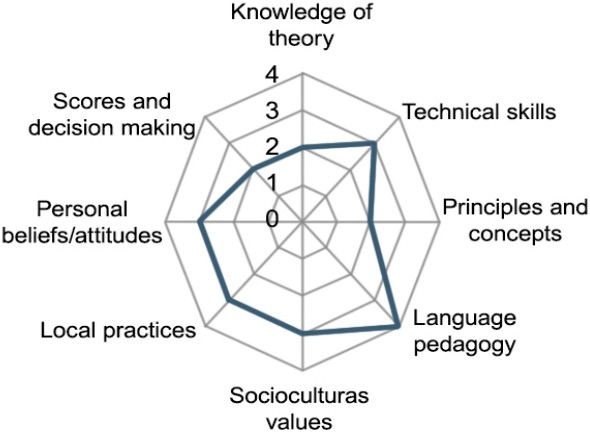

**Figure 1.** Taylor's (2013) LAL profile.

According to this model, LAL aspects that are most important for language teachers are language pedagogy, while scores and decision making, knowledge of theory and principles and concepts are least important. Bøhn and Tsagari (2021) explained the specified dimensions of a language teacher LAL construct by the fact that teacher LAL has a strong focus on "teaching-related aspects" (p. 229). Similarly, Scarino (2013) stressed that assessment is indispensable for the curriculum and processes of teaching, and thus learning and assessment in schooling should be aligned. Farhady (2018) reiterated the idea of the necessity to match the current understanding of language learning and use with the related assessment theory and practice to be able to meet different cultural and language realities. However, some studies yielded results on quite similar LAL concerns in different educational contexts. For example, participants in Hasselgreen et al. (2004) and in Vogt and Tsagari's (2014) surveys displayed similar low levels of LAL and a need for more tangible guidance in self-, peer and alternative assessment. We can assume that, on the one hand, LAL has recently become a universally recognized part of every teacher's professional competence. Yet, the shift from assessment of learning to assessment for learning is just becoming a global trend and requirement in education settings. In this light, Coombe et al. (2020) concluded that assessment for learning has become a more dominant theme among other modern themes of assessment standards in the USA from 2010 until the present. However, understanding educational and cultural landscapes and teacher beliefs seems to contribute to a more accurate specification of LAL training levels and needs (Vogt et al. 2020).

Among other studies that support that LAL has a contextually local rather than universal nature is the study by (Bøhn and Tsagari 2021; see also Drackert and Wolfgang 2018; Xu and Brown 2016; Hakim 2015; Hasselgreen et al. 2004), which was conducted through a constructed interview in order to investigate Norwegian teacher educators' perception of the LAL construct as well as the relevance of Taylor's (2013) model for the Norwegian context. The results reveal that Norwegian teachers' perception of the LAL model aligned to a great degree with the dimensions of Taylor's model. Yet, teacher educators in Bøhn and Tsagari's study provided an even broader conceptualization of Taylor's model (the refined model now includes 10 dimensions) with a much stronger focus on teaching-related aspects (Bøhn and Tsagari 2021), which the authors explained to be the result of Norwegian educational policies. Norwegian teachers perceive disciplinary knowledge, collaboration competence, principles and concepts, local practices and scores and decision making as very relevant; thus, their perception of the LAL construct includes more skills and knowledge (Figure 2).

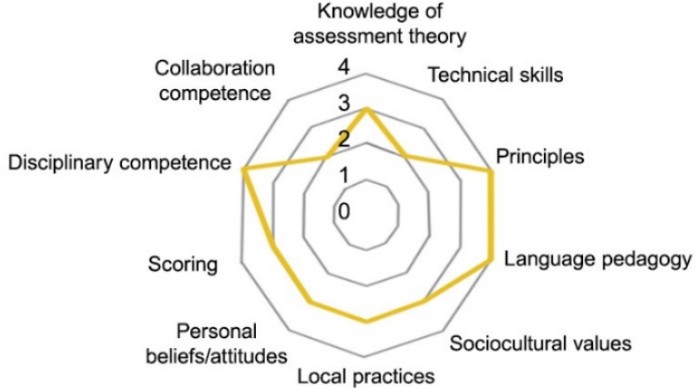

**Figure 2.** Bøhn and Tsagari's refined LAL profile.

In order to understand Ukrainian teachers' LAL needs better, we focused more specifically on the study of writing assessment literacy of Ukrainian university language teachers (Kvasova and Kavytska 2014; Kvasova et al. 2019; Kvasova and Shovkovy 2020). The results obtained from 104 tertiary English teachers provided insight into local current teachers' needs in the assessment of writing (Kvasova et al. 2019). The findings show that

the approach to students' writing assessment is often intuitive as assessment in Ukraine is not regulated by common standards. This may explain the generally insufficient level of teachers' writing assessment literacy. As a result, teachers struggled with the purpose of assessment writing or with the choice of authentic writing tasks. In addition, the prevailing majority of teachers confessed they could not tell the difference between holistic and analytic scales, though many claimed they applied scales while rating. Moreover, teachers reported to lack practical skills related to scoring and interpretation of assessment results. Consequently, data obtained indicated that learning-oriented assessment practices are not commonly employed to foster learning. Thus, it was found that feedback is not yet a meaningful interaction between a Ukrainian teacher and a learner. Similar findings were elicited through the study of teaching in Ukrainian tertiary schools conducted by the British Council in Ukraine (Dexter 2019), which reported concerns about the lack of student-centered practices in teaching overall and the lack of methods employing assessment for learning in particular.

To develop a workshop construct that focuses on perceived training needs in writing assessment literacy, we matched Ukrainian university teachers' concerns with the dimensions of the refined model of LAL (Bøhn and Tsagari 2021) and adopted clarification of knowledge and skills that should be developed and/or enhanced during training (Table 1).

**Table 1.** Workshop construct according to Ukrainian teachers' concerns in writing assessment literacy.

| Ukrainian Teachers' Concerns | LAL Refined Model by Bøhn and Tsagari | Clarification of Workshop Goals |
| --- | --- | --- |
| Why and how to assess writing concerns | Disciplinary competence | Knowledge of low- and high-level skills and the cognitive processes utilized in writing. Knowledge of authentic writing tasks at tertiary level. |
| Intuitive approach to assessment | Language pedagogy | Differentiating between assessment of, for and as learning. |
| No understanding of a common standard | Knowledge of theory + Local practices | Knowledge of assessment theory. Knowledge of curriculum. |
| Difficulties in differentiating between holistic and analytic scales | Scoring | Knowledge of scoring processes and use of rating scales. |
| Low employment of assessment for learning | Principles and concepts | Developing an understanding of how learning can be promoted. |
| Scoring concerns and the need for developing an understanding of how to interpret results | Language pedagogy | Enhancing skills of communicating learning goals, assessment criteria and the ability to provide good feedback. |

Table 1 demonstrates the writing assessment literacy needs of Ukrainian language teachers that, in our opinion, should outline the goals of a workshop. These needs mainly emphasize five dimensions of a Ukrainian teacher LAL profile: language pedagogy; scoring; knowledge of assessment theory; principles; and disciplinary competence.

Thus far, we have considered how mentoring training can be organized within an institution in order to achieve desirable outcomes.

The research questions were:

(1) Will mentoring training, conducted within a short (1-month) period, enable inexperienced presenters to develop and conduct an effective workshop in LTA?
(2) In what way(s) has mentoring training impacted the mentees as prospective trainers in LTA?

## 4. Materials and Methods

### 4.1. The Purpose of the Mentoring and Participants

The training purpose was to organize a 1-day dissemination event "Assessment of writing in universities" within an Erasmus+ program at Vasyl Stus Donetsk National University. Other objectives were to guide the beginning presenters through the initial professional stage in LTA to improved professional development. Tangible evidence of the mentees' professional growth is correlated with their ability to design and conduct a workshop in LTA that meets LAL needs of Ukrainian university teachers. At the end of the training, each mentee conducted a one-hour workshop, followed by a 15 min discussion. For these purposes, informal mentoring training that simulated the standard framework of formal training was carried out (Table 1). The mentoring training was conducted at Taras Shevchenko National University of Kyiv.

The study examined its research questions by analyzing the data collected from two sets of participants. The first set consisted of two mentees, teachers of Taras Shevchenko National University of Kyiv with basic presenters' skills but no experience in the development or conduct of dissemination events. Both mentees received adequate training in LTA as a result of their participation in Erasmus+ staff mobility programs. Participation in the training was voluntary and happened naturally. The dissemination event in the form of a workshop in LTA was a part of their commitment to Erasmus+ staff mobility program. The training lasted for one month and was conducted by one mentor, who was an expert in LTA, the founder and president of UALTA and a grant holder of several international projects in LTA. The mentor had received formal training on how to prepare trainers by participating in various related programs.

The second set ($n = 37$) represented attendees of the workshop, university teachers of Vasyl Stus Donetsk National University. Their participation was voluntary and was based on their professional interest and needs in LTA. All participants were informed about the purpose of the study and the use of their data. Informed oral consent was obtained from all individual participants included in the study.

### 4.2. Mentoring Design

Developing a conceptual framework specific to the professional development of the novice presenters allowed us to consolidate our beliefs and align them with the theory and research in the field (Table 2). Mentoring toward professional growth in the field of LTA and developing positive self-efficacy employs varied aspects of training, as discussed earlier in the article. The training goals and activities are explicitly linked to improving the needs of the mentees and to the outcomes of the training.

The mentoring training design adopted flexible approach to the instruction and included multiple forms of mentoring in order to increase motivation and achievement. For example, the participants of the training experienced group mentoring mode, in which like-minded people shared common interests and outcome expectations. They met to discuss important field-related issues, such as the construct of the workshop, Ukrainian university English teachers' LAL concerns, etc. In addition, mentees met with a mentor in a one-on-one mode to discuss their drafts and practice workshop delivery.

Table 3 illustrates the stages of the mentoring training that consisted of three face-to-face meetings, mentees' individual research and filling in the reflection logs.

**Table 2.** Summary of specifications of the mentoring training.

| Specifications of the Mentoring Training | Descriptors of the Training |
|---|---|
| 1. The training goal is related both to the mentees' professional and psychological growth. | - Academic motivation and research inquiry in the field of LTA; <br> - Promotion of professional self-efficacy through recognition of one's professional growth and overall skills improvement (performance, behavioral and cognitive) to produce desirable results (to develop a workshop to promote university English teachers' LAL). |
| 2. The training design is based on research in mentee's induction and development and effective mentoring practices. | - Self-efficacy theory; <br> - Social cognitive theory; <br> - Social cognitive career theory; <br> - Adult development theory; <br> - Productive mentoring. |
| 3. The training expectation outcome. | - Design an effective workshop that meets (country) university English teachers' needs in LTA (Table 1). |
| 4. The training is based on standards that promote interpersonal relationship and self-regulatory skills. | - Goal orientation; <br> - Cooperation; <br> - Common interests; <br> - Mutual respect; <br> - Support and guidance; <br> - Crossover efficacy; <br> - Physical proximity; <br> - Frequency of meeting; <br> - Reflection on and analysis of the experience. |

**Table 3.** The procedure of the mentoring training.

| | | |
|---|---|---|
| 1 | Self-reflection stage <br> Developing LAL construct stage | - Filling in "before the start of meeting 1" reflection log; <br> - Literature analysis in order to define the construct of the workshop. |
| 2 | Meeting 1 | - Establishing a trusting and positive relationship; <br> - Outlining objectives and outcomes of the workshop; <br> - Defining workshop framework (based on the results of the research stage); <br> - Understanding trainees' interests and strong points. |
| 3 | Home assignment <br> (a week) | - Related literature investigation; <br> - First draft preparation. |
| 4 | Meeting 2 | - Delivery of the 1st drafts; <br> - Discussing strong and weak points. |
| 5 | Home assignment (a week) | - Revisiting and editing the presentation drafts. |
| 6 | Meeting 3 | - Delivery of the 2nd drafts; <br> - Discussing strong and weak points. |
| 7 | Home assignments (a week) | - Minor presentation draft editing; <br> - Preparation of the presentation summary and workshop handouts. |
| 8 | Gauging self-efficacy | - Determining mentees' perceived efficacy. |
| 9 | Workshop | - Delivering a dissemination workshop at Vasyl Stus Donetsk National University. |
| 10 | Data collection stage | - Gauging workshop success; <br> - Filling in "after the workshop" reflection log. |

The mentoring was designed to give all the participants of the training the possibility to plan and develop the workshop in LTA, reflect upon strong and weak sides of the workshop and grow professionally in LTA.

Before the first meeting, the trainees were provided with a thoroughly developed procedure of the training and the reflective trainees' logs, some parts of which they were expected to fill in already before the meeting; the other part was completed after the dissemination event. "Before the meeting" reflection logs aimed at preparing the trainees for a more productive cooperation. They were asked to outline their expectations of the training, the role of the mentor in their professional development and the amount of time they were ready to work every day to successfully fulfill the objectives of the training.

In addition, the trainees conducted analysis of the related literature on language teachers' current LAL needs in order to develop the construct of the workshop training. The results of the research presented in Table 1 were finalized together with the mentor during the first meeting.

During the first meeting, a trusting, positive and working relationship between the supervisor and the novice presenters was reinforced through negotiating views of the workshop and sharing ideas. The meeting also set up the objectives and expected outcomes of the upcoming workshop. Moreover, this "initial meeting" provided the supervisor with insight, in the most general sense, into the trainees' competence in the field of LTA and their ideas about what they thought could be included in upcoming workshop. For example, during the meeting, the mentor found out that one of the trainees opted for the part of the workshop that could be characterized as more theoretically grounded, while the other volunteered to prepare a more practical part of the workshop. In addition, the novice presenters received a home task to analyze related literature and prepare first drafts of their parts of the workshop. By the end of the week, the trainees did considerable inroads into literature investigation, revisited questions and presentation objectives discussed during the meeting and prepared their first presentation drafts.

During the second meeting, the mentees piloted their workshops under real-time settings. The mentor and colleagues-volunteers performed the parts of workshop attendees. After that, strong and weak points of the workshops were discussed, context-based questions asked, feedback received and new guidelines outlined. During the following week, the presentation drafts were revisited and edited. The final versions of the workshops were presented at the 3rd meeting. It should be also noted that the mentees had the opportunity to seek their mentor's support outside the meetings as well. Regular calls and emails were a part of mentoring training.

*4.3. Measures*

The findings of the study conducted by Creswell (2012) demonstrate that characteristics of a good mentoring program are observable. A mixed-methods research study that includes various data collection methods such as interviews, focus groups, reflection diaries/logs and surveys can provide an in-depth understanding of what constitutes a successful mentoring program.

A qualitative method built on the interpretative phenomenological analysis (IPA) model (Morrow 2007; Smith et al. 2009; Whitehouse 2016) was used to explain the data of the reflection logs in order to understand mentees' lived experiences in the mentoring training.

Besides the qualitative data, quantitative methods that tested both research questions 1 and 2 were employed. A paper–pen questionnaire determined the success of the workshop among the attendees of the workshop (see Appendix A Table A1 for the results of the questionnaire survey) and a self-efficacy scale determined the mentees' perceived efficacy (see Appendix B).

4.3.1. Qualitative Method

To test research question 2, reflection logs were used (see Appendix C). Data of the reflection logs were analyzed using IPA model introduced by Smith et al. (2009). This

model "invites participants to offer a rich, detailed, first person account of their experiences" (Smith et al. 2009, p. 56). The logs consisted of 11 structured questions, generated on the basis of a review of the mentoring literature and were focused around four main themes:

1.   Realized expectations;
2.   Quality time with mentor;
3.   Mentor–mentee relations;
4.   The impact of the training.

The reflection logs included two parts. The first part was filled in before the training to outline mentees' expectations and set personal goals for the training. The idea to complete reflective logs before the start of the training is based on the evidence that it is possible to foresee mentees' perception of mentoring by recognizing their constructs at the outset of the program (Blake 2016). The second part was filled in after the workshop in LTA was conducted with the purpose to understand the success of the training and the context that pertained to the intricacies of mentor–mentee relations. The information of the reflection logs was later used in analysis and description of mentoring experience.

### 4.3.2. Quantitative Methods

The questionnaire was disseminated immediately after the workshop among the attendees of the workshop ($n = 37$). Participants were told that the questionnaire was designed to better understand whether mentoring could be viewed as an effective training tool to organize workshops in LTA. All answers were anonymous to encourage the participants to share their true perceptions of the workshop. Responses were made by rating the workshop using a 10-point scale, with 10 being excellent. The attendees were asked to identify workshop quality by answering three questions: whether (1) attendees gained new knowledge and broadened their understanding of LTA; (2) the workshop was worth attending; and (3) the workshop objectives were achieved. In addition, the participants measured the presenters' organizational skills by rating how the workshop was run and whether it was engaging. It took the attendees around 15 min to answer the questionnaire. Paper–pen responses were collected by the mentor.

The mentees' perceived self-efficacy level was gauged before the workshop in LTA with the help of 8 "can do" items that reflected the construct of the training outcomes. Responses were scored on a 5-point scale that ranged from 0 (cannot do) to 4 (highly certain can do). Higher scores indicated higher level of self-efficacy.

Additionally, data on mentor–mentee interaction frequency were collected. The frequency was operationalized by determining the average number of hours spent in face-to-face meetings or other forms of communication (calls and emails, etc.) as indicated by the participants of the study.

### 4.3.3. Analysis

The reflection logs analysis was conducted through descriptive and conceptual interpretative lenses (Smith et al. 2009). First, reflection logs were analyzed to identify patterns and ideas clustered around main themes. Then, it was important to go beyond mere description and produce deeper interpretative analysis by drawing on the comparative analysis of the findings in related studies. Finally, the conclusions were made with the aim to introduce the distinctive opinion of the participants about mentoring rather than existing notions in related literature.

Descriptive data analysis method was employed to interpret the findings collected through the questionnaire and self-efficacy scales. The measures of central tendency (mean and mode) and measures of variability (range and interquartile range (IQR)) were examined to discuss the research questions.

## 5. Results

*5.1. Results for RQ 1*

Research question 1 asked whether mentoring training, conducted within a short (1-month) period, will enable inexperienced presenters to develop and conduct an effective workshop in LTA?

Table 4 shows the means and ranges of the workshop success level among 37 workshop attendees (see Appendix B for descriptive results).

**Table 4.** The results of the survey gauging workshop success level on a scale from 1 to 10, with 10 being excellent.

| Survey Questions | Mean | Mode | Range | IQR |
|---|---|---|---|---|
| Q1. Was the workshop in LTA run smoothly and on time? | 9.4 | 9 | 1 | 1 |
| Q2. How can you assess the presenters' skills in making sure the workshop was engaging and interesting? | 8.8 | 9 | 3 | 2 |
| Q3. How much more knowledge or understanding in LTA have you gained from the workshop in comparison to before you started? | 7.7 | 8 | 2 | 1.5 |
| Q4. Was attending workshop in LTA worth your time invested in the meeting? | 8.5 | 8 | 3 | 2 |
| Q5. How well was the goal of the workshop in LTA achieved today? | 8.4 | 8 | 3 | 2 |

The results reveal that the short-term mentoring training yielded positive results. The participants' general perception of the success of the workshop scored higher than 7; the most frequent score regarding questions 1 and 2 was 9 and regarding questions 3, 4, 5–8, which speaks about the overall success of the training. The data also report a low variability of both range and interquartile range that indicate and confirm the consistency of the collected results.

*5.2. Results for RQ 2*

Research question 2 asked in what way(s) mentoring training has impacted the mentees as prospective trainers in LTA?

The result of the self-efficacy scale (see Table 5) reported that structured short (1-month) mentoring contributed significantly to raising the mentees' level of self-efficacy. It is worth noting that the mentees demonstrated high certainty about statements 2, 5, 7 and 8 and a bit lower degree of confidence about statements 1, 3, 4 and 6. This can be explained by that the former set of statements was entirely related to the pre-prepared part of the workshop, while the second set of the statements questioned the mentees' ability to maintain flexibility to change and efficacy in spontaneous situations. Naturally, the level of efficacy in situations that may not be fully predicted and foreseen improves with experience which takes time.

The reflection logs (see Appendix C) helped to examine the relations between the expectations and outcomes of the mentoring training. The collected data of Part A of the reflection logs demonstrated that the mentees expressed their hopes that mentoring would primarily provide them with the strategies of how to design and conduct workshops in LTA.

**Table 5.** Results of the perceived self-efficacy on the scale from 0 to 4, where 0—Cannot do at all; 1—Not sure can do; 2—Moderately can do; 3—Can do; 4—Highly certain can do.

| | Self-Efficacy Survey | Mentee 1 | Mentee 2 | Mean |
|---|---|---|---|---|
| 1 | Get the audience's attention at the beginning of the workshop | 3 | 3 | 3 |
| 2 | Deliver the workshop clearly and logically | 4 | 3 | 3.5 |
| 3 | Be confident during the workshop | 3 | 2 | 2.5 |
| 4 | Be flexible if necessary | 3 | 2 | 2.5 |
| 5 | Feel confident about the topic | 4 | 4 | 4 |
| 6 | Respond to the audience's needs | 3 | 3 | 3 |
| 7 | Finish the workshop effectively | 4 | 3 | 3.5 |
| 8 | Answer the audience's questions | 4 | 4 | 4 |

Clearly, the mentees associated the quality of the training, at least partially, with the mentor's personality and her expertise in the career field: "*I think, the mentor relationship is one of the crucial factors in mentoring either enabling maximizing mentees' effort into successful performance or discouraging their effort.*" (Mentee 1)

In addition, the atmosphere of collaboration, discussion, mutual planning and research helped to maintain a unifying atmosphere of team building and collective responsibility. "*Together with the help in designing the workshop in LTA, the mentor provided us with some useful tips on time management of the workshop, incorporating interactive activities into the workshop to encourage workshop participation and ways to successfully wrap up the presentation. Moreover, the mentor coupled all her non-judgmental and constructive comments after our rehearsal presentations with practical advice on how to remedy the issues regarding either editing their content or improving them technically, which have definitely met all the expectations and even more.*" (Mentee 2)

In this particular experience, it should be noted that the mentees and the mentor did not have to overcome the phase of unease, pass the stage of accommodating to one another or develop a trustful atmosphere. The mentor had long gained the respect of every member of a research community for her ability to be supportive of beginner colleagues. "*Happily, there was no a stage, where the mentor and mentees had to accommodate to one another as long as we have been colleagues for a long time, having the experience of participating in conferences, seminars and workshops, coauthoring articles, etc.*" (Mentee 2)

"Emotional closeness" of that particular cooperation was mentioned by the mentees and contributed to the success of the training. "*Some of the initial fear and discomfort of non-recognition went quickly away as the result of the mentor's skillful and clear guidance as well as her supportive attitude*" (Mentee 1). Thus, effective mentorship is not generally all about career outcomes but more about seeing mentorship within a profession as a relationship rather than a set of duties (Awaya et al. 2003). Quality mentoring as reported by mentees in many studies pertains to the development of emotional closeness, provides the sense of mutual interdependence and provides psychosocial guidance (Allen et al. 2006; Ragins et al. 2000).

The present study also provided evidence that physical proximity that enabled "*regular face-to-face meetings*" and "*the possibility to pilot workshops*" was an important factor for the successful outcome of the training. However, the results regarding time spent in training are mixed.

Part B of the reflection log was completed after the workshop was conducted. The analysis of the mentees' responses showed that they had to cope with many challenges connected to overcoming the fear of not meeting the mentor's expectations, boosting their own confidence, managing schedules and planning and editing the workshop.

On the one hand, the time spent in the face-to-face meetings that allowed the mentees to share and discuss as well as pilot and rehearse their presentations directly correlated with the mentees' perceived success of the training outcome. "*Due to her practical guidance and literally time invested into mentoring, I feel more confident and ready to present at the*

*workshop.*" (Mentee 2). On the other hand, the research evidence proves that more hours in mentoring training are negatively associated with mentorship quality, career training and role modeling (Allen et al. 2006). Allen et al. explained that investing more time into mentoring might be regarded by mentors as time intrusion or more time invested into mentoring might exceedingly raise mentors' expectations of the program outcomes. In this regard, Blake (2016) suggested that it might be necessary for organizations where mentoring is not a formal professional practice to "recognize and allow time for mentorship within the professional's job description" (Blake 2016, p. 10) in order not to discourage volunteer mentoring for formal schemes. Unfortunately, informal mentoring in Ukrainian context is most commonly viewed as a senior and more knowledgeable friend's support rather than professional guidance. In addition, we did not collect the mentor's responses about the mentoring training in this study, which deprived us of the possibility to analyze the mentor's point of view.

Adjusting schedules and setting regular times might be a real challenge that, unfortunately, cannot be always resolved efficiently. According to common research evidence, it is usually mentees who express dissatisfaction with the fact that their mentors are often unavailable for in-person meetings (Whitehouse 2016). This was not the case in this study. Although all the participants were involved in day-work during the working week, the mentor found time for them. This sacrifice on the part of the mentor was acknowledged with the expression of concern: "*The biggest unease for me was to know that our mentor-mentee meeting schedule made the mentor wait for the mentees to get free from their work, but I feel it didn't go in vain;) And after the seminar feedback proves it!*" (Mentee 2).

Tailoring a mentor's sets of skills to meet their protégés' needs might be another challenge if training should be mutually beneficial (Whitehouse 2016). It is reported that trainees' previous experience of collaboration or wrong expectations of the mentor's role in building their professional development can negatively impact the training process (Blake 2016). In this regard, we believe, the challenge was at least partially overcome by completing "before the training" reflection logs that helped the mentees more accurately specify and understand the mentor–mentee roles and each other's responsibilities: "*I expect to build up self-confidence and self-efficacy, and strengthen the effectiveness of my presentation through careful planning, organization, and before the workshop presentation practice.*" (Mentee 1); "*I expect that my mentor can guide me in finding the objective of the workshop regarding LAT, help in outlining the workshop presentation, choosing proper supporting materials and tailoring an effective workshop in reference to the needs of the participants, and to give her constructive feedback on the clarity and effectiveness of my rehearsal presentation.*" (Mentee 1); "*By consistently following the mentor's guidelines and completing all the assignments, listen carefully to comments to be able to edit the initial draft of the presentation in terms of topics or skills to cover, assigning an estimated length of time for each item on the plan, the pace of delivering, etc.*" (Mentee 2); "*I think, this must be a shared responsibility involving contribution and consistency from both sides.*" (Mentee 2). The experience of the mentees in this study supports the proposition by Blake (2016) that without reflective development the effects of mentoring can significantly regress. Obviously, reflection logs are enormously effective in sorting out mixed emotions, expectation outcomes and promoting a more conscious, goal-oriented approach to any activity.

Another finding in terms of challenges for that particular experience was related to fears of "*not knowing the level of expertise of the workshop participants*" (Mentee 2). In this experience, the mentor suggested preparing additional content for more experienced attendees. Thus, particular handouts for participants with higher expertise were prepared for them to stay involved throughout the workshop. Naturally, meeting one's audience's professional needs and their level of expertise and being able to build one's presentation on what the audience knows and not just repeat things they know are key ingredients of any successful and effective workshop. Keeping one's audience in mind while designing a workshop must be an important feature of a trainer's identity.

In addition, the results demonstrate that the mentees described many benefits of the received training. The most significant finding is the crossover effects between the results of the self-efficacy scale and perceived benefits of the mentoring training expressed in the reflection logs. It shows a correlation between the mentees' perceived self-efficacy level and the quality of the training. From the mentees' perspective, the perceived mentor–mentee input into the training became critical to the success of the workshop in LTA. "*I am absolutely happy with time spent with the mentor as long as I feel all my expectations related to delivering my presentation at the workshop were met*" (Mentee 1). To some extent, this result is similar to the finding reported by Allen et al., which indicated that "mentors with protégés who perceived that they had greater input into the match reported greater mentorship quality than did mentors with protégés who perceived that they had less input into the match" (Allen et al. 2006, p. 574).

Data analysis of the mentor–mentee interaction frequency showed that the actual time spent in face-to-face meetings by far exceeded the mentees' initial intentions of daily commitment of around 1.5 h, as in-person meetings lasted up to three hours (rehearsal time included). Thus, we can assume that a short mentoring intervention brings positive outcomes if it is characterized by a high mentor–mentee interaction frequency.

## 6. Discussion

The present study examined the possibilities of mentoring training used as a platform to coach inexperienced presenters to develop and conduct an effective workshop in LTA. Furthermore, this study analyzed the effects mentoring training had on the mentees as prospective trainers in LTA. The results reveal several correlations between the level of the workshop success and mentorship quality.

The structured strategy of the mentoring program positively correlated with the expectations about the outcomes of the training. The findings of this study reiterated the idea that establishing a reliable and caring but working relationship is a key ingredient of successful mentor–mentee cooperation (Hofmann and Springer 2014; Whitehouse 2016). The participants reported their professional development in the field of LTA and improvements in practical skills in organizing and conducting workshops. On the global level, the participants' understanding of the LAL profile for teachers changed from a universally conceptualized model to a culturally specific one. In order to be a trainer in LTA, penetration into local practices and understanding cause and effect for teachers' beliefs in a particular environment are significant. Specifically, the mentees stated that research into Ukrainian university English teachers' needs helped them identify concerns of the particular cohort. Understanding teachers' needs resulted in accurate conceptualization of the workshop design and tailoring the workshop construct that focused on teachers' current concerns in LTA.

The expected outcome of the mentoring training was the successful delivery of the workshop in LTA which, in the mentees' expectation, could be achieved if the mentor helped to plan the workshop, provided guidance and constructive feedback, participated in editing and provided strategies of effective delivery. Moreover, it was important for the mentees to be able to discuss and share their ideas without being judged. The mentees acknowledged that the mentor's explicit approval of mentees' efforts encouraged and motivated them to think outside the box and beyond the basics. In mentees' perception, positive effects of face-to-face communication with the mentor could be maximized by completing some assignments in advance before the meeting. This, they believed, could help them understand their weak points, which could be included in the agenda of the following meeting. The success of mentoring, in the mentees' view, requires their personal commitment and perseverance that could be measured, for example, by time devoted to the training and consistency of the effort. They also viewed the mentor–mentee relation as shared responsibility, collaborative interchange and learning.

In addition, the mentees' responses demonstrated that the clear-cut structure of the mentoring and the specified construct of the workshop, as well as guidance and the

mentor's tangible input into the training, made the process highly engaging and motivating. Supportive, nonjudgmental guidance increased confidence and provided a sense of self-efficacy. As a result, both mentees admitted that they expressed professional confidence in undertaking the training in LTA as one of their career responsibilities. They also specified that the positive outcomes of the workshop helped them understand they possessed necessary qualities, the identity of a trainer and knowledge in LTA. This finding indicates that mentoring helped mentees identify their capacity as prospective trainers. The mentees admitted that undertaking mentoring as a career commitment requires constant personal and professional development in LTA. They also believed that to be able to train other professionals effectively, it is essential to receive outside support and the possibility to upgrade in LTA and training strategies. Thus, professional trainers' development needs an intelligent and sustainable view.

The conclusion aligns with the developmental model of self (Bachkirova 2011). According to the model, the evolution of the sense of self is influenced by the interaction with many factors while passing through four stages of development: identity strength, unformed ego, formed ego and reformed ego. In terms of a mentor's development, a similar idea was echoed by Merrick and Stokes (2003), who suggested that, in their progress, mentors pass four stages of learning. During the first three stages, a mentor needs extensive and varied outside support and guidance in order to move from an initial to an advanced stage.

The findings have some limitations because of a single data source. The findings are built exclusively on the mentees' responses. As a result, the mentor's opinion about interaction frequency, mentorship and the perceived outcomes of the training was not collected. Since "mentors and protégés may use different criteria in assessing mentorship quality" (Allen et al. 2006, p. 576), the findings provide only a partial understanding of the ability of training to prepare inexperienced presenters to develop and conduct an effective workshop in LTA. Moreover, the study did not present a mentor's point of view about how training impacted the prospective trainers in LTA. One more limitation is the sample of the participants of the study ($n = 2$). Although it has been suggested that, in IPA research, "there is no right answer to the question of . . . sample size" (Smith et al. 2009, p. 56) and even a single participant study could be justified for a convincing case, we cannot argue that our findings are representative for drawing consistent recommendations and conclusions.

Another limitation is that informal mentorship is inevitably biased considering that it is commonly based on mutual likability. In this experience, the mentor–mentee relationship was characterized by long-standing cooperation and support beyond professional obligations. This fact might have influenced the mentees' responses, which otherwise could have been different, for example, in a formal mentoring scheme. Nevertheless, in our view, informal mentoring, which is fueled by emotional closeness and supported by similarities of views, can be a powerful tool for professional growth and achievement of desirable targets.

The success of the mentoring training was empirically tested through a single dissemination event. However, more advanced research in other departments and institutions is necessary in order to verify the positive effects of the described mentoring framework.

## 7. Conclusions

Here we described our experience of mentoring from the mentees' perspective to better understand the advantages and challenges of mentoring conducted with the purpose of assisting less-experienced professionals during organizing and conducting a dissemination event in LTA. The informal mentoring training based on a close, intense and, hopefully, mutually beneficial relationship was structured and carried out in agreement with the findings on effective mentoring described in the literature. The mentoring framework was developed according to formal standards that proved to be effective strategies to carry out study objectives. The participants reported improvement in their organizational skills

and professional growth as a result of the practical opportunities the training provided and the mentor's guidance and assistance. The outcome of that well-intended training brought positive returns in the form of a successful workshop, the mentor's recognition and the mentees' improved professional position. In addition, mentoring helped the mentees recognize their trainers' potential. In Ukrainian context, mentoring at tertiary level is not a common or formal practice. Consequently, informal mentoring is hardly ever recognized as one of the ways of a purposeful process to foster professional development and improve career growth. The implication may include the change of attitude to informal mentoring as an effective tool of providing career support to novices in their professional field. As a result, an organization might consider schemes to attract motivated experts to "reinforce desired norms and values" (Blake 2016, p. 3). In this study, a flexible approach was used to mentoring instruction to better respond to the participants' needs. Further research focus might be on developing a mentoring training which is based on mentees' needs analysis in order to identify trainees' profiles more accurately. A study aimed at understanding mentoring framework characteristics to sustain the transition of a novice to expert mentor in LTA might be of research interest. Our next step, though, is exploring a mentoring framework that will help prepare trainers for conducting online courses in LTA.

**Author Contributions:** Conceptualization, O.K. and V.O.; methodology, V.O. and O.D.; software, O.D.; validation, V.O., O.D. and O.K.; formal analysis, V.O.; investigation, O.K.; resources, V.O.; data curation, O.K.; writing—original draft preparation, V.O.; writing—review and editing, O.D.; visualization, O.D.; supervision, O.K.; project administration, O.K. All authors have read and agreed to the published version of the manuscript.

**Funding:** This research received no external funding.

**Institutional Review Board Statement:** Institutional review board approval statement was issued on 24 September 2021, Approval number 044/93-26.

**Informed Consent Statement:** Not applicable.

**Data Availability Statement:** The study does not contain any generated data on any public available database. All data is provided in the study. The paper-pen questionnaire results and reflection logs results are archived together with other research paper at home.

**Conflicts of Interest:** The authors declare no conflict of interest.

## Appendix A

**Table A1.** Results from the questionnaire survey gauging workshop success level on a scale from 1 to 10, with 10 being excellent.

| | Participants (*n* = 37) | | | | | | | | | | | | | | | | | | | | | | | | | | | | | | | | | | | | |
|---|---|---|---|---|---|---|---|---|---|---|---|---|---|---|---|---|---|---|---|---|---|---|---|---|---|---|---|---|---|---|---|---|---|---|---|---|---|
| | **1** | **2** | **3** | **4** | **5** | **6** | **7** | **8** | **9** | **10** | **11** | **12** | **13** | **14** | **15** | **16** | **15** | **18** | **19** | **20** | **21** | **22** | **23** | **24** | **25** | **26** | **27** | **28** | **29** | **30** | **31** | **32** | **33** | **34** | **35** | **36** | **37** |
| Q1 | 10 | 10 | 9 | 9 | 9 | 9 | 9 | 9 | 10 | 10 | 10 | 9 | 9 | 9 | 9 | 10 | 10 | 9 | 9 | 9 | 10 | 10 | 10 | 10 | 10 | 9 | 9 | 10 | 10 | 9 | 10 | 9 | 10 | 9 | 10 | 9 | 9 |
| Q2 | 10 | 9 | 8 | 10 | 9 | 8 | 8 | 7 | 8 | 8 | 9 | 9 | 9 | 9 | 10 | 10 | 9 | 10 | 7 | 9 | 9 | 8 | 8 | 10 | 10 | 9 | 8 | 9 | 9 | 7 | 9 | 8 | 8 | 10 | 10 | 9 | 9 |
| Q3 | 9 | 8 | 8 | 8 | 8 | 7 | 7 | 7 | 8 | 7 | 8 | 8 | 8 | 8 | 8 | 8 | 7 | 7 | 7 | 8 | 8 | 9 | 7 | 8 | 8 | 7 | 8 | 8 | 7 | 7 | 8 | 8 | 7 | 8 | 8 | 8 | 7 |
| Q4 | 10 | 8 | 8 | 8 | 8 | 9 | 9 | 10 | 10 | 8 | 8 | 7 | 7 | 8 | 9 | 9 | 9 | 8 | 8 | 8 | 10 | 10 | 9 | 9 | 9 | 8 | 8 | 8 | 7 | 8 | 9 | 9 | 8 | 8 | 9 | 9 | 9 |
| Q5 | 10 | 8 | 8 | 8 | 8 | 8 | 9 | 10 | 9 | 8 | 8 | 8 | 7 | 8 | 9 | 8 | 9 | 8 | 8 | 8 | 9 | 10 | 10 | 9 | 9 | 8 | 8 | 8 | 8 | 8 | 9 | 8 | 8 | 8 | 9 | 9 | 9 |

## Appendix B

**Table A2. Self-efficacy scale:** Rate your degree of confidence by recording a number from 0 to 4 using the scale given below.

| | | | **Cannot Do at All** | **Not Sure Can Do** | **Moderately Can Do** | **Can Do** | **Highly Certain Can Do** |
|---|---|---|---|---|---|---|---|
| | | | **0** | **1** | **2** | **3** | **4** |
| | 1 | Get the audience's attention at the beginning of the workshop | | | | | |
| | 2 | Deliver the workshop clearly and logically | | | | | |
| | 3 | Be confident during the workshop | | | | | |
| | 4 | Be flexible if necessary | | | | | |
| | 5 | Feel confident about the topic | | | | | |
| | 6 | Respond to the audience's needs | | | | | |
| | 7 | Finish the workshop effectively | | | | | |
| | 8 | Answer the audience's questions | | | | | |

**Appendix C**

Reflection log

Part A: Before the initial meeting reflection log:

What do you expect from the training?

What do you expect of your mentor?

How much time do you expect to devote to your training every day?

How do you get the most out of your time with the mentor?

How do you determine your responsibility in relation to the responsibility of the mentor?

Part B: After training reflection log:

What expectations of your mentoring relationship have been met?

Are you happy with your mentor's input into the training?

Was time spent with the mentor satisfactory?

How did mentor relationship enhance or influence workshop organization and performance?

What challenges were experienced during mentoring and how were they overcome?

Do you recognize training in LTA as part of your prospective responsibility?

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
