# Peer review of "Training the Trainers in Language Assessment via Mentoring: Building Expertise to Promote Language Assessment Literacy of Ukrainian University Teachers"

_languages, doi:10.3390/languages6040194_

Round 1

Reviewer 1 Report

The paper takes up a very promising and to date underresearched area of language assessment literacy (LAL) and of ways how to improve it, namely mentoring. It relates the background, method and findings of an explorative small-scale qualitative study of one mentor and two mentees.

The literature review comprises a part on LAL and a part on mentoring. The part on LAL is comprehensive and timely and uses relevant and recent literature. The part on mentoring often draws on the same selection of literature and would benefit from a little more scope. Also, it is lacking a theoretical model or framework for mentoring that could form the basis of the subsequent empirical study while the LAL part of the workshop is well theorised.

While the procedure of the mentoring training is adequately presented in Table 1, more contextual information is needed e.g. about the activities and outcome of the workshop (What kind of presentation is required from the participants?) and what exactly happened (and why) in the mentoring sessions (e.g. one-to-one mentoring? One-to-many? Typical procedures? How is the mentoring linked with the workshop?)

In terms of method, a more systematic imparting of the relevant information would be helpful: What was the sample? N=37 participants or N=2 mentees? How was the sample generated? Study design: It seems this was a mixed methods study rather than a qualitative one. The questionnaire and self-assessment rating scale were described but not mentioned as data collection instruments. No information is given on data analysis methods. In the research questions, the role of LTA is not too clear. Is it the goal of the intervention to develop participants’ LTA or are they to be enabled, by means of the intervention, to conduct workshops in order to develop their / other people’s LTA or both? This should be clarified (RQ2) both in the methods and in the results section.

The findings are clearly structured along the RQs but would profit from a little more detail, e.g. when participants reported to have developed their professionalism, what exactly did they say they developed? You might also want to link this back to the concept of LTA, which is central to the study. Otherwise the findings related remain on the surface.

In the discussion section, the author(s) introduce new findings from reading logs and relate them to relevant literature. Rather than calling this content discussion, it should be made part of the results chapter and integrated in the same structure along the RQs rather than just relating the answers. Answers from the questionnaires are not taken up at all in this part.

The conclusion is brief, giving a brief summary of the study results and implications. The part on future research avenues could be expanded a little.

There are some language glitches and some formal flaws, e.g. in the reference section which does not follow the stylesheet.

Author Response

Response to Reviewer 1 Comments

Comment 1.

The literature review comprises a part on LAL and a part on mentoring. The part on LAL is comprehensive and timely and uses relevant and recent literature. The part on mentoring often draws on the same selection of literature and would benefit from a little more scope. Also, it is lacking a theoretical model or framework for mentoring that could form the basis of the subsequent empirical study while the LAL part of the workshop is well theorised.

Response 1: Theoretical framework for the mentoring training is added in section 2.2.1 (lines 159-188).

Comment 2.

While the procedure of the mentoring training is adequately presented in Table 1, more contextual information is needed e.g. about the activities and outcome of the workshop (What kind of presentation is required from the participants?) and what exactly happened (and why) in the mentoring sessions (e.g. one-to-one mentoring? One-to-many? Typical procedures? How is the mentoring linked with the workshop?)

Response 2: The explanation and the details about the activities and outcome of the workshop, the purpose and mode of the workshop are provided (lines 321-329 – the purpose and outcome of the training; section 4.2 gives an insight into mentoring design; lines 356-364 explain activities and the mode of mentoring).

Comment 3.

In terms of method, a more systematic imparting of the relevant information would be helpful: What was the sample? N=37 participants or N=2 mentees? How was the sample generated?

Response 3: The information about two sets of participants clarified (see lines 330-345).

Comment 4.

Study design: It seems this was a mixed methods study rather than a qualitative one. The questionnaire and self-assessment rating scale were described but not mentioned as data collection instruments. No information is given on data analysis methods.

Response 4. The study used a mixed methods analysis to interpret the collected data. This information was included into the measures section. The details of the mixed methods analysis were provided. The information about the data analysis methods was also added (see lines 456-467).

Comment 5.

In the research questions, the role of LTA is not too clear. Is it the goal of the intervention to develop participants’ LTA or are they to be enabled, by means of the intervention, to conduct workshops in order to develop their / other people’s LTA or both? This should be clarified (RQ2) both in the methods and in the results section.

Response 5: We added the explanation about the purpose of the mentoring training in the material and methods section that, in our opinion, clarifies RQ 2. A modified results section that includes now the reflection logs findings also makes RQ2 clear and explicit.

Comment 6.

The findings are clearly structured along the RQs but would profit from a little more detail, e.g. when participants reported to have developed their professionalism, what exactly did they say they developed? You might also want to link this back to the concept of LTA, which is central to the study. Otherwise the findings related remain on the surface.

Response 6: The explanation to the mentees’ professionalism development is added (see line 625-633).

Comment 7.

In the discussion section, the author(s) introduce new findings from reading logs and relate them to relevant literature. Rather than calling this content discussion, it should be made part of the results chapter and integrated in the same structure along the RQs rather than just relating the answers. Answers from the questionnaires are not taken up at all in this part.

Response 7: The reading logs findings were removed from the discussion section and integrated in the structure of the RQ analysis section.

Comment 8.

The conclusion is brief, giving a brief summary of the study results and implications. The part on future research avenues could be expanded a little.

Response 8: The part on future research was expanded, see lines 713-720.

Comment 9.

There are some language glitches and some formal flaws, e.g. in the reference section which does not follow the stylesheet.

Response 9: The article was proofread for language glitches. Such problems as misuse / absence of articles, lengthy sentences, punctuations, spelling mistakes, etc. were attended to.

The reference section was reviewed in accordance with the requirements of the stylesheet.

Reviewer 2 Report

Dear authors

This is an interesting study on training the teacher trainers in language teaching assessment via mentoring. Your article,  based on sound methodology which is also clearly and sufficiently described, offers valuable insights to those interested in applying your mentoring model in other countries and contexts. Thank you for enlightening me in this respect.

I have marked language problems, suggesting corrections, on the manuscript, please see in attachment.

Author Response

Response to Reviewer 2 Comments

Comment 1

I have marked language problems, suggesting corrections, on the manuscript, please see in attachment.

Response 1: The authors gratefully accepted all the suggestions on corrections to the manuscript and attended every language and other problems that the Reviewer marked.

Such problems as misuse / absence of the articles, lengthy sentences, punctuations and spelling mistakes, misuse of the genitive case, problems of the subject-predicate agreement, etc. were attended to.

The missing sources were also added to the reference section.

The range of the questionnaire responses was calculated and this information was included in table 4.

The Figures were represented in the text in accordance with the Reviewer’s suggestion.